# An Update on Thiol Signaling: S-Nitrosothiols, Hydrogen Sulfide and a Putative Role for Thionitrous Acid

**DOI:** 10.3390/antiox9030225

**Published:** 2020-03-10

**Authors:** Nadzeya Marozkina, Benjamin Gaston

**Affiliations:** 1Herman Wells Center for Pediatric Research, Riley Hospital for Children, School of Medicine, Indiana University, Indianapolis, IN 46202, USA; begaston@iu.edu; 2Indiana University, School of Medicine, 1044 W. Walnut Street, R4-474 Indianapolis, IN 46202, USA

**Keywords:** S-nitrosothiols, hydrogen sulfide, thionitrous acid, antioxidants

## Abstract

Long considered vital to antioxidant defenses, thiol chemistry has more recently been recognized to be of fundamental importance to cell signaling. S-nitrosothiols—such as S-nitrosoglutathione (GSNO)—and hydrogen sulfide (H_2_S) are physiologic signaling thiols that are regulated enzymatically. Current evidence suggests that they modify target protein function primarily through post-translational modifications. GSNO is made by NOS and other metalloproteins; H_2_S by metabolism of cysteine, homocysteine and cystathionine precursors. GSNO generally acts independently of NO generation and has a variety of gene regulatory, immune modulator, vascular, respiratory and neuronal effects. Some of this physiology is shared with H_2_S, though the mechanisms differ. Recent evidence also suggests that molecules resulting from reactions between GSNO and H_2_S, such as thionitrous acid (HSNO), could also have a role in physiology. Taken together, these data suggest important new potential targets for thiol-based drug development.

## 1. Introduction

Thiol chemistry has recently been recognized to be important for the regulation of cell signaling processes. S-nitrosoglutathione (GSNO) and other S-nitrosothiols, as well as hydrogen sulfide (H_2_S), are enzymatically-regulated cell signaling thiols, affecting target protein function through post-translational modifications. Regulation of the metabolism of GSNO, of other S-nitrosothiols and of H_2_S (Figure 1) are vital to immune, vascular, respiratory and neuronal signaling. Here, we will review data regarding the role and regulation of these molecules in physiology.

## 2. Overview of S-Nitrosothiol Metabolism 

### 2.1. Production of S-Nitrosothiols and Other Nitrogen Oxides In Vivo 

Nitric oxide synthases (NOS) use the guanidino nitrogen of l-arginine [1,2,3,4] to form NO, and GSNO is also often formed when NOS is activated [5,6,7,8]. Three principal isoforms of NOS exist [9,10,11]. These are neuronal (nNOS), endothelial NOS (eNOS) and inducible NOS (iNOS). Activation of all three can result in downstream S-nitrosothiol formation [6,9,10,12] (Figure 1). Reactions of NO with thiols are typically kinetically favored to occur through interactions with metalloproteins rather than through inorganic NO oxidation. Typically, they involve NO oxidation to NO^+^. NO is oxidized by reactions with superoxide and with certain metal ions. NO also reacts with oxygen, but this reaction is third order (third order rate constant ~7 × 10^3^ L^2^ mol^−2^ sec^−1^) [13]: The reaction is normally quite slow in physiology because the concentrations of NO in the body gas phase are ~nM. The reaction rate is favored in membranes [14] because the reactants are more soluble in lipid than water. As discussed more below, GSNO and other S-nitrosothiols are typically formed through reactions of NO with iron and copper groups in these metalloproteins (including NOSs), though there is evidence also for formation of a complex with an amine of NOS-bound tetrahydrobiopterin (BH_4_) [15] and other reactions [16]. Other metalloproteins involved in S-nitrosothiol formation include Hb, where S-nitrosylation occurs at the b-93 cysteine [17], and ceruloplasmin [18]. 

Endothelial NOS provides an example of the role of S-nitrosothiol formation and NO in biology [19,20,21]. We and others have shown that eNOS activation can result in S-nitrosothiol formation [12,22]. Moreover, GSNO catabolism by GSNO reductase is required to reverse and or modulate the effects of eNOS, suggesting that, in physiology, GSNO is a regulated eNOS product [19,23]. At the myoendothelial junction, eNOS is colocalized with somatic cell hemoglobin alpha [20]. When the iron of the hemoglobin is oxidized, eNOS activation produces NO and S-nitrosothiols; when it is reduced, inert nitrate is formed [22]. Note that impaired activity of eNOS resulting from oxidative stress in the endothelium involves BH_4_ oxidation [24,25]. This induces the uncoupling of eNOS, with consequent generation of O_2_^−^ instead of NO [24,26] or GSNO [12]. Endothelial NOS redox chemistry, with products including thiol-stabilized S-nitrosothiols, inert nitrite and nitrate and highly ONOOH, are relevant to a range of vascular diseases [25,26,27,28,29,30,31,32,33,34,35,36] 

### 2.2. S-Nitrosylation Signaling and S-Nitrosothiols 

Broadly, thiols modified by a NO^+^ (RS^−^ - NO^+^), NO^−^ (R-S^+^-NO^−^) or NO (RS-NO) are referred to as S-nitrosothiols [27]. The thiol R group can be a simple proton (thionitrous acid, see below), or can be an amino acid, peptide or protein. S-Nitrosylation is a normally a targeted, post-translational protein cysteine modification with a specific biological purpose. Note that S-nitrosylated proteins may transfer NO to other proteins [28,29,30]. S-nitrosylation is regulated as a signaling reaction. [5,12,31,32]. Regulation of human airway tone provides an example. GSNO is normally present in the human airway at ~500 nM (30), the IC_50_ for soluble guanylate cyclase (sGC)-independent human airway dilatation [33]. GSNO prevents β2 agonist tachyphylaxis by S-nitrosylating GRK2 downstream of eNOS activation through its effect on β arrestin to [19]. Indeed, tachyphylaxis to β2-AR agonists is ablated in [19,34] mice lacking the ability to reduce GSNO because of deletion of the enzyme, GSNO reductase (GSNOR). These mice are protected from β-AR [34], and humans with gain-of-function in this enzyme are at increased asthma risk [35].

There are many additional examples of signaling caused by GSNO and other S-nitrosothiols [5,6,12,31,32,36]. These molecules regulate a broad range of bioactivities, ranging from coagulation and N-methyl-d-aspartate (NMDA) dependent neuronal signaling [37,38]. In many cases, NO^+^ is transferred from a low-mass S-nitrosothiol to a protein thiol. A recent example involves S-nitrosylation of connexin 43 (Cx43) hemichannels in the Duchenne Muscular Dystrophy (Dmd) heart that has been driven by β2 agonists to develop stress-induced arrhythmias: inhibition of NOS prevents arrhythmias evoked by isoproterenol [39]. NOS activation results in S-nitrosylation of Cx43 at cysteine 271, regulating arrhythmia risk. [39]. 

Other mechanisms of thiol modification are relevant as well. Myeloperoxidase causes protein thiol S-nitrosylation at tyrosine-Xn-cysteine (YXnC) motifs in the presence of hydrogen peroxide and nitrite [16]. Inorganic S-nitrosylation can occur at low pH, particularly in the mitochondrial intermembrane space [40], in the lung and gut, and in the blood vessels in ischemic tissues [41,42]. In the case of hemoglobin- and ceruloplasmin-mediated S-nitrosylation, NO radical normally first reduces the metal: Fe^3+^-NO transitions to Fe^2+^-NO^+^; Cu^2+-^ NO transitions to Cu^+^-NO^+^. Of note, GSNO can serve as an intermediate to transfer the NO^+^ equivalent from a metalloprotein to a target thiolate. 

Note that the GSNO breakdown product, S-nitroso-L-cysteine (L-CSNO), is increasingly appreciated as a stereoselective signaling molecule. L-CSNO was originally considered an endothelium derived relaxing factor (EDRF) [43,44,45,46]. Lewis, Bates and others observed that the L-isomers of CSNO and related S-nitrosothiols were substantially more active than the D-isomers, though they evolve NO equally [47,48,49]. Their findings suggest that L-CSNO activates stereoselective recognition sites [50]. In guinea-pig myocytes, L-CSNO activates large conductance Ca^2+^ activated K+-channels by S-nitrosylation of the alpha subunit [51]. In porcine coronary arteries, L-CSNO activates intermediate and small conductance Ca^2+^-activated K^+^-channels. S-nitroso-D-cysteine (D-CSNO) does not have these effects [45,46]. Ca^2+^-activated K^+^-channels activation causes vasorelaxation exclusively from voltage-gated Ca^2+^ channel closures induced by hyperpolarization [52]. Though nifedipine, a voltage-gated Ca^2+^ +channel blocker, reduces vasodilator responses from donors of NO, it does not reduce the responses of L-CSNO or the release of EDRF by acetylcholine [52]. 

Signaling mediated by S-nitrosothiols is not unique to mammalian physiology: it is vital in plant biology and in microbiology. Interaction between eukaryotic host and its microbiome has been shown to be affected by inter-species S-nitrosothiol signaling [53]. For example, prokaryotic S-nitrosothiol signaling uses the hybrid-cluster protein Hcp downstream of nitrate reductase. Here, the complex interactions of S-nitrosothiol synthases and transnitrosylases signal *E. Coli* mobility and metabolism [36].

### 2.3. Denitrosylation

Like other post-translational modifications, S-nitrosylation signaling is balanced by the reverse effect, denitrosylation. Many enzymes function as denitrosylases. Commonly, these use NAD(P)H oxidation to reduce S-nitrosothiols. Examples include NADH-dependent GSNO reductase (see below) and NADPH-dependent SNO-Coenzyme A reductase (Figure 1). Many additional enzyme systems catabolize GSNO, the details of which are beyond the scope of this review [54,55,56,57,58,59,60] Products can include another S-nitrosothiols, NO, ONOOH, NH_2_OH and NH_3_. SNO-CoA reductase is also known as aldo-keto reductase family 1 member A1 (AKR1A1) (Figure 1). AKR1A1 can catabolize both SNO-CoA and GSNO [61]. 

These S-nitrosothiol catabolic enzymes play critical roles in human physiology. For example, GSNOR gene variants affect asthma prevalence and β2 agonist responsiveness in specific human asthma subpopulations [35,62,63]; we have identified a specific severe asthmatic population phenotype associated with high GSNOR activity [64]. Denitrosylation functions are vital, as evidenced by functional redundancy. For example, GSNOR ^-/-^ mice have more AKR1A1 GSNO catabolic activity than do wild type mice [61]. 

## 3. Hydrogen Sulfide Production and Metabolism In Vivo. Similarities and Differences of NO and H_2_S Metabolism

Like GSNO, H_2_S is a thiol-based mediator that is produced enzymatically and is active in physiology at low, endogenous concentrations [65,66,67]. Hydrogen sulfide is produced by mammalian enzymes cystathione-lyase(CSE), cystathione-synthase (CBS) and 3-mercaptopyruvate sulfurtransferase (3-MST) [68]. Additionally, cysteine aminotransferase (CAT) produces 3-mercaptopyruvate substrate for 3MST to produce a protein-bound persulfide, that in turn can be reduced to form H_2_S [69,70] (Figure 1), and, in plants, by desulfhydrases [71]. Note that CBS and CSE are multifunctional enzymes catalyzing a range of β replacement and α, β and γ elimination reactions: Their roles in thiol chemistry can often be redundant (Figure 1). The substrate for 3MST is 3-mercaptopyruvate; others can use L-cysteine, cystathione and other substrates (Figure 1).

As with S-nitrosothiol-based signaling, H_2_S can affect physiology by causing post-translational modifications of protein thiols. Also as with S-nitrosothiols, specific enzymes also break down H_2_S [72]. For example, mitochondrial rhodanese, a sulfur transferase, catalyzes H_2_S oxidation [73], as a component of a major H_2_S catabolic pathway that also involves a sulfide quinone oxido-reductase (SQR) and a sulfur dioxygenase. H_2_S catabolic enzymes are extensively reviewed in ref [74]. Additionally, iron-containing proteins, such as hemoglobin and cytochrome C oxidase, also can serve as sinks for H_2_S (71,77). 

The first physiological protein target identified for H_2_S involved vasodilatation mediated by H_2_S modification of the K _ATP_ (Kir6.x) channel [75]. Like S-nitrosothiols, H_2_S also acts through post-translational modifications of protein. The key reaction is defined as S-sulfhydration [76,77], the forming protein –SSH moieties. This S-sulfhydration reactions can modify a large number of proteins, including GAPDH (at Cys150), albumin, actin, ADH1, AST, catalase and thioredoxin-reductase [76]. Indeed, there could well be cross-talk between S-nitrosylation and S-sulfhydration signaling [78].

Inhibition of NOS by the nonspecific inhibitor L-NAME leads to inhibition of H2S-induced vasodilation [79,80,81], while the deletion of CSE (cystathionine γ-lyase) enhances the vasodilatory effects of acetylcholine [82]. 

Like S-nitrosothiols, H_2_S signaling has a role in regulating vascular tone. CSE^−/−^ mice have increased blood pressure as they age [83]. Isolated vessels from CSE^−/−^ mice have significantly impaired methacholine-induced vasodilatation. Of note, activation of endothelial CSE and eNOS are both Ca^2+-^calmodulin dependent [83,84]; and both can involve cGMP and phosphodiesterases, with competing effects. [85,86].

In contrast to some of the more labile S-nitrosothiols; however, H_2_S is relatively stable in certain body compartments. However, it can be metabolized by methylation or oxidation, and its products excreted in urine [87,88]. It is also is scavenged by its reactions with heme. H_2_S for example, because of its interaction with mitochondrial cytochrome c oxidase [67,89], can be toxic at higher concentrations [90,91]. Two H_2_S oxidative pathways have been identified recently and involve cytosolic ferric (met) hemoglobin (Hb) in red blood cells [92] and myoglobin (Mb) in the heart [93]. Both of these proteins have the capability to generate heme-bound polysulfides and free thiosulfates as end-products. Decreasing pH (across the range 8–6.8) decreases heme H_2_S-affinity by causing a right-shift in the binding of H_2_S to metHb. Equilibrium dissociation constants of metHb and H_2_S are in the range of 0.26–1.08 μM, which is well below that for NO it (~100 μM) [94] and CN−(~10 μM) [95], among other metHb ligands. This indicated that H_2_S is a high-affinity metHb ligand. Rate constants for metHb (3.2 × 10^3M^^−1s^^−1^) [37] and metMb (1.6 × 10^4M^^−1s^^−1^) [96] suggest that metHb is a reversible H_2_S carrier, affecting H_2_S bioactivities [95].

### Thionitrous Acid and Related Compounds

Thionitrous acid (HSNO) is another low molecular weight thiol that may be relevant to signaling. HSNO can be formed by reactions between hydrogen sulfide (H_2_S) and S-nitrosothiols [97,98,99]. For example, HSNO is produced rapidly according to, H_2_S + GSNO → HSNO + GSH. Theoretically, HSNO can also be formed by the reaction N_2_O_3_ → HSNO + HNO_2_. However, as noted above, N_2_O_3_ is not formed abundantly in most physiological systems. 

It has been shown that the NaHS/GSNO reaction products relax precontracted rat aortic rings with pharmacology consistent with the relevance of HSNO or SNO-intermediates. These are dependent on heme concentrations, as well as pH [100]. Of note, there is evidence that HSNO can diffuse through membranes to signal in physiology [98], supporting a possible role as a signaling molecule. Note, however, that the pKa of HSNO is low, and HSNO in physiology may exist as the SNO^−^ anion. A molecule, QT490, has been developed that forms HSNO and provides insight into formation of HSNO/RSNO from the reaction between H_2_S/RSH and NO in the biological system [101].

It is also possible that HNO and HSSH could be formed from HSNO with H_2_S. The resulting S–N bond is the longest reported S–N bond for an R-SNO compound, 1.84 Å [102]. The tautomeric form of HSNO is more thermodynamically feasible and kinetically accessible [103]. These reactions could also form bioactive nitroxyl HNO [103]. HNO is a powerful vasodilator. While NO reacts slowly with thiols, the reaction of HNO with thiols is very fast and it depends on the pKa of the reacting thiolate [104]. Rapid HNO production can also occur in small-to-medium-sized sensory neurons and axons, where it was observed to be co-expression of CBS (cystathionine beta synthase) with TRPA1 (transient receptor potential channel A1) [79]. TRPA1 is the Ca^2+^ ion channel responsible for CGRP release induced by HNO), [79] which is known to be co-expressed with nNOS. Reactive cysteine residues on TRPA1 become oxidized by HNO, resulting in long-lasting channel opening, Ca^2+^ influx, and subsequent release of CGRP [105]. These three proteins, CBS, nNOS, and TRPA1, represent a functional unit that is involved in the regulation of peripheral blood flow [106] and even the regulation of systemic blood pressure [79] It has been hypothesized that the disturbances in the regulation of this pathway might be responsible for the migraine attacks [106]. 

Taken together, these data suggest the possible relevance of HSNO at the interface of S-nitrosothiol and H_2_S signaling.

## 4. Conclusions

Small thiol molecules have cell signaling effects in physiological concentrations. Both S-nitrosothiols and H_2_S can be formed and broken down by specific regulatory proteins. Both have physiological effects involving post-translational modifications, primarily at cysteine residues, but the mechanisms are different. Once considered primarily valuable for breaking disulfide bods to reduce viscosity and/or to augment antioxidant defenses, thiol chemistry is beginning to be recognized as relevant to cell signaling reactions. These recent observations suggest that thiol chemistry may be more useful than previously thought for new drug development.

However, more work needs to be done to understand in greater details the roles of S-nitrosothiols and of H_2_S in physiology, and to determine the degree to which HSNO chemistry is relevant to biology.

## Figures and Tables

**Figure 1 antioxidants-09-00225-f001:**
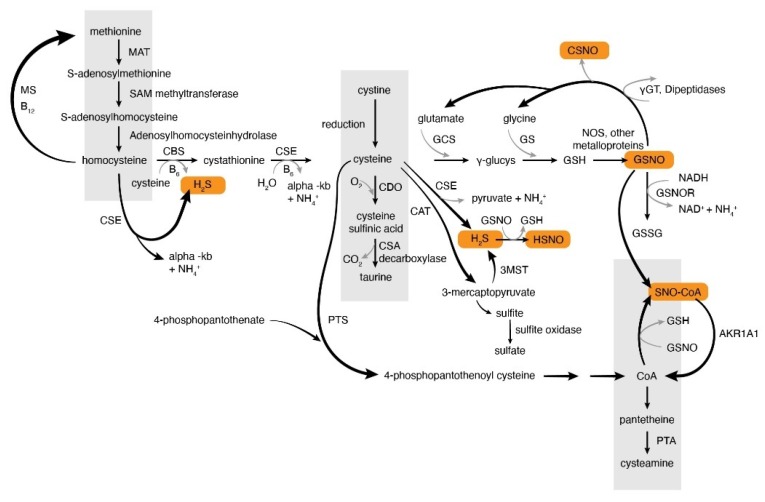
Eukaryotic thiol chemistry and the generation of thiol-based signaling molecules. Orange highlight shows product signaling molecules. AKR1A1—aldoketoreductase family 1 member 1A, CAT—cysteine aspartate aminotransferase, CBS—cystathionine b-synthase, CSE—cystathionine gamma lyase, CDO—cysteine deoxygenase, CoA—Coenzyme A, C-SNO—S-nitrosocysteine, GSNO—S-nitrosoglutathione, GSNOR—S-nitrosoglutathione reductase, GS—glutathione synthase, GCS—glutamylcysteine synthase, MAT—methionine adenosyltransferase; SAM- S-adenosyl methionine; 3MST—3 mercaptopuruvate sulfur transferase, PTA—pantetheinase, SNO-CoA—S-nitroso-Coenzyme A, γGT—γ-glutamyl transpeptidase.

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
