# Peer review of "An Update on Thiol Signaling: S-Nitrosothiols, Hydrogen Sulfide and a Putative Role for Thionitrous Acid"

_antioxidants, 2020, doi:10.3390/antiox9030225_

Round 1

Reviewer 1 Report

This manuscript intends to cover significant aspects of thiol signaling and does touch the recent findings. However it does so superficially and lacks precision in statements and sounds, at parts, negligent. I started to read the manuscript with enthusiasm but somehow disappointed by the lack of thoroughness in mechanistic aspects of thiol signaling.

I will provide a few examples of problematic statements but it is really a problem throughout the manuscript.

1. The sentence in pg-3 line 109 reads “Indeed, interaction between the microbiome and the eukaryotic host has been shown to be mediated, at least in part, by inter-species S-nitrosothiol signaling”. This is unsettling for a biological process that is too complex. Perhaps what it meant is “inter-species S-nitrosothiol signaling also play a role in microbiome and the eukaryotic host interaction.

2. Similarly paragraph starting at line 42 sounds like NOS produces nitrosothiol species. NOS produce NO, which further reacts to form nitrosothiols and these reactions are not catalyzed by NOS. Metal dependent mechanisms are not limited to NOS enzymes. The same is stated line 133.

3. Line 136. It is stated that CAT produces H2S. CAT does not produce H2S. It makes 3-mercaptopyruvate which is used by 3MST to produce a protein bound persulfide that in turn leads to H2S formation in the presence of a reductant. The study cited here is about the role of CBS in malignant hypothermia. It is important to cite original articles.

4. Similarly the statement in line 142 “specific enzymes also remove H2S moieties” deserves detail and citation of the original work and by itself is problematic.

5. The paragraph starting in line 147 about H2S oxidation is ignorant and misleading. Authors left out the main mitochondrial oxidation pathway for H2S but highlighted some high abundant iron dependent proteins, which function as sinks for H2S and produce polysulfides, as the only oxidation pathways for H2S. This is problematic.

Author Response

Reviewer 1

  1. The sentence in pg-3 line 109 reads “Indeed, interaction between the microbiome and the eukaryotic host has been shown to be mediated, at least in part, by inter-species S-nitrosothiol signaling”. This is unsettling for a biological process that is too complex. Perhaps what it meant is “inter-species S-nitrosothiol signaling also play a role in microbiome and the eukaryotic host interaction.

Answer.  We agree and we have rephrased this sentence. Please see p 3, line 112-114.

  1. Similarly paragraph starting at line 42 sounds like NOS produces nitrosothiol species. NOS produce NO, which further reacts to form nitrosothiols and these reactions are not catalyzed by NOS. Metal dependent mechanisms are not limited to NOS enzymes. The same is stated line 133.

Answer.  Thank you for this suggestion.  We have changed this section to emphasize NO oxidation as an intermediate to S-nitrosothiol formation; and we have further emphasized the metalloproteins can be involved in NO oxidation.

  1. Line 136. It is stated that CAT produces H2S. CAT does not produce H2S. It makes 3-mercaptopyruvate which is used by 3MST to produce a protein bound persulfide that in turn leads to H2S formation in the presence of a reductant. The study cited here is about the role of CBS in malignant hypothermia. It is important to cite original articles.

Answer.  We agreed that CAT (cysteine aspartate aminotransferase) does not directly produce H2S, but rather coupled with mercaptopyruvate sulfurtransferase in the subsequent reaction. This was in our figure, but we have clarified the intermediates in the text (p 4, line 139-142). The study we cited there, in regard to the role of Cystathionine beta-synthase in malignant hyperthermia, reference 72, is an original article. We also have added another reference (71) showing the role of 3-MST and CAT in production of H2S.   

  1. Similarly the statement in line 142 “specific enzymes also remove H2S moieties” deserves detail and citation of the original work and by itself is problematic.

Answer. Thank you for this suggestion.  We have expended this section, please see p 4, line 146 – 150, and cited more original articles (ref 75, 76)

  1. The paragraph starting in line 147 about H2S oxidation is ignorant and misleading. Authors left out the main mitochondrial oxidation pathway for H2S but highlighted some high abundant iron dependent proteins, which function as sinks for H2S and produce polysulfides, as the only oxidation pathways for H2S. This is problematic.

Answer. Thank you for your comment. We have discussed to a greater extent the  main mitochondrial oxidation pathway enzymes for H2S oxidation, and have de-emphasized the iron-dependent proteins. Please see p 4, line 146- 153.

Reviewer 2 Report

This is a nice brief update about S-nitrosothiols and H2S cross-talk. Since HSNO is included even in the title, I would advise the authors to expand a little bit the last section. Some contributions in this field have been made recently but are not mentioned here. The authors should look at Ivanovic-Burmazovic & Filipovic, Inorg Chem, 2019 for detailed overview, as well as Kang et al, Angew Chem Int Ed, 2018 and Islam et al, Inorg Chem, 2017 for examples of potential HSNO donors for cells.

Author Response

Reviewer 2

  1. Since HSNO is included even in the title, I would advise the authors to expand a little bit the last section. Some contributions in this field have been made recently but are not mentioned here. The authors should look at Ivanovic-Burmazovic & Filipovic, Inorg Chem, 2019 for detailed overview, as well as Kang et al, Angew Chem Int Ed, 2018 and Islam et al, Inorg Chem, 2017 for examples of potential HSNO donors for cells.

Answer. Thank you for your comment. We have expanded the last section of HSNO chapter to incorporate these suggestions (please see p 5, line 191 – 195, 200 – 211) and included recommended references (ref 81, 82, 83, 84, 103, 104, 107, 108, 109).

Reviewer 3 Report

In this review manuscript, the authors described the roles of S-nitrosothiols, H2S, and thionitrous acid in regulating cellular functions via different mechanisms. The authors concluded that these molecules maybe used for new drug development. I listed some comments as shown below:

The English in this manuscript is hard to understand and may need to be improved.

More discussion on metalloproteins involved in S-nitrosothiols can be further expanded with a summarized figure. Are NOSs metalloproteins, as indicated by the authors in line 54?

More clear evidence for the generation of S-nitrosothiols from eNOS need to be provided and further described.

As discussed for the roles of S-Nitrosylation and denitrosylation from S-nitrosothiols, the signals of H2S in posttranslational modification of proteins by sulfhydration and desulfhydration need to be discussed as well.

The future direction and possible problem for researching these thiol compounds can be provided in the last paragraph.

Author Response

Reviewer 3

  1. The English in this manuscript is hard to understand and may need to be improved.

Answer. We have made major stylistic edits throughout.

  1. More discussion on metalloproteins involved in S-nitrosothiols can be further expanded with a summarized figure. Are NOSs metalloproteins, as indicated by the authors in line 54?

Answer. Yes, all 3 NOS (eNOS, iNOS and nNOS) are metalloproteins with heme centers that affect nitrogen redox state. We have expanded the discussion of other metalloproteins. We have expanded this information into the review. Please see p 2, line 54-58.

  1. More clear evidence for the generation of S-nitrosothiols from eNOS need to be provided and further described.

Answer. Thank you for this suggestion.  This is further expanded on page 2-3. We and others have shown that eNOS activation can form S-nitrosothiols (for example, Straub Nature [24] and Gow JBC ([2]).    However, we think it is also important to note that some of the most compelling evidence is that GSNOR is often required to counter-regulate the effects of eNOS (for example, Whalen Cell [21]).  That is, the effect of eNOS is reversed and/or modulated specifically by GSNOR, suggesting that decreasing GSNO decreases the effect of eNOS.  This discussion is now expanded.

  1. As discussed for the roles of S-Nitrosylation and denitrosylation from S-nitrosothiols, the signals of H2S in posttranslational modification of proteins by sulfhydration and desulfhydration need to be discussed as well.

Answer. We have included additional information with regard to H2S posttranslational modification of proteins by sulfhydration (please see p 4, line 155-158) and desulfhydration (please see p 4, line 139-141)

  1. The future direction and possible problem for researching these thiol compounds can be provided in the last paragraph.

Answer. Thank you for your comment. We have expanded the paragraph about future directions in thiol signaling in the concluding paragraph.

Round 2

Reviewer 1 Report

The revised manuscript addresses the concernes I had. 

Reviewer 3 Report

The authors have properly addressed my comments and I would suggest it to be accepted at this version.

This manuscript is a resubmission of an earlier submission. The following is a list of the peer review reports and author responses from that submission.